# Vibrational Spectroscopy as a Sensitive Probe for the Chemistry of Intra-Phase Bacterial Growth

**DOI:** 10.3390/s20123452

**Published:** 2020-06-18

**Authors:** Kamila Kochan, Elizabeth Lai, Zack Richardson, Cara Nethercott, Anton Y. Peleg, Philip Heraud, Bayden R. Wood

**Affiliations:** 1Centre for Biospectroscopy and School of Chemistry, Clayton Campus, Monash University, Clayton, VIC 3800, Australia; ellai2@student.monash.edu (E.L.); zack.richardson@monash.edu (Z.R.); phil.heraud@monash.edu (P.H.); 2Infection and Immunity Program, Monash Biomedicine Discovery Institute and Department of Microbiology, Clayton Campus, Monash University, Clayton, VIC 3800, Australia; cara.nethercott@gmail.com (C.N.); anton.peleg@monash.edu (A.Y.P.); 3Department of Infectious Diseases, The Alfred Hospital and Central Clinical School, Monash University, Melbourne, VIC 3004, Australia

**Keywords:** growth phase, lag phase, ATR, Raman, *S. aureus*

## Abstract

Bacterial growth in batch cultures occurs in four phases (lag, exponential/log, stationary and death phase) that differ distinctly in number of different bacteria, biochemistry and physiology. Knowledge regarding the growth phase and its kinetics is essential for bacterial research, especially in taxonomic identification and monitoring drug interactions. However, the conventional methods by which to assess microbial growth are based only on cell counting or optical density, without any insight into the biochemistry of cells or processes. Both Raman and Fourier transform infrared (FTIR) spectroscopy have shown potential to determine the chemical changes occurring between different bacterial growth phases. Here, we extend the application of spectroscopy and for the first time combine both Raman and FTIR microscopy in a multimodal approach to detect changes in the chemical compositions of bacteria within the same phase (intra-phase). We found a number of spectral markers associated with nucleic acids (IR: 964, 1082, 1215 cm^−1^; RS: 785, 1483 cm^−1^), carbohydrates (IR: 1035 cm^−1^; RS: 1047 cm^−1^) and proteins (1394 cm^−1^, amide II) reflecting not only inter-, but also intra-phase changes in bacterial chemistry. Principal component analysis performed simultaneously on FTIR and Raman spectra enabled a clear-cut, time-dependent discrimination between intra-lag phase bacteria probed every 30 min. This demonstrates the unique capability of multimodal vibrational spectroscopy to probe the chemistry of bacterial growth even at the intra-phase level, which is particularly important for the lag phase, where low bacterial numbers limit conventional analytical approaches.

## 1. Introduction

Bacterial growth in batch cultures typically occurs in four distinct phases that are clearly visible through the growth curve: (1) lag, (2) exponential (log), (3) stationary and (4) death phase [1,2,3]. Each phase is associated with specific characteristics with respect to growth rates, reflecting events occurring in each phase that are associated with chemical changes [1,2,3]. The initial lag phase is characterized by cellular activity that is not yet followed by bacterial division. Within the lag phase, bacterial cells undergo adaptation to new environmental conditions and prepare for subsequent growth [2]. This is associated with synthesis of a variety of compounds and is reflected in the growth of the sizes of existing cells [4]. Subsequently, the cells enter the log phase, in which the exponential growth occurs at a constant rate, with cells multiplying though binary fission [1,2,3]. Eventually, bacterial populations reach a stationary phase in which conditions become unfavorable (e.g., due to depletion of nutrients and a progressive accumulation of waste), resulting in a decline of population growth. Cell growth reaches a plateau when the number of dividing cells equals the number of dying cells. Finally, in the death phase the number of living cells decreases and the population growth declines rapidly [1,2,3].

Knowledge of the bacterial growth phase, bacterial numbers and growth kinetics is essential for research and commercial applications. Conventional methods of determining the growth phase are based on cell numbers, which are established either through standard plate counting or through optical density [5]. These methods, however, cannot describe any biochemical changes, which would reflect microbial physiology. This is of particular importance in context of the lag phase, the physiology of which remains elusive [2]. The low concentrations of bacteria in the lag phase are an additional challenge for practical use of genomic approaches in research [2].

Over the past decade, both Raman and FTIR spectroscopy have become widely used in microbiology research on fungi [6,7,8,9] and bacteria [10,11,12,13,14,15]. FTIR spectroscopy is commonly applied for bacterial classification, with the ability to reach strain-level discrimination. Numerous reviews are available in the literature, providing an overview of this application [10,13,14]. IR in its conventional application enables the study of bacteria on a population level because the diffraction limit of IR light prevents a resolution from being achieved that is adequate for single bacterial cells. This limitation was recently overcome by combining atomic force microscopy with IR spectroscopy (AFM-IR) and probing the photothermal expansion resulting from IR absorption [12]. The application of AFM-IR for bacterial studies in various contexts was demonstrated [11,12,15]. Raman spectroscopy is also rapidly gaining popularity in bacterial studies, with numerous examples of strain-level discrimination on single cells [16,17,18,19].

Both IR and Raman were also applied to study various aspects of growth [20,21,22,23,24,25,26,27]. The strong influence of growth conditions and the composition of medium on the biochemistry of bacteria was demonstrated at a population level by FTIR [20] and at the single-cell level using Raman microspectroscopy [21]. FTIR was also used to demonstrate changes in the overall chemical composition of bacterial populations during growth, with a specific focus on inter-phase differences [22,23]. Similar studies have been conducted on single cells by means of Raman spectroscopy [24,25,26]. However, hitherto no reports have focused primarily on the study of biochemical changes within individual growth phases. This is particularly important for the lag phase. Both Raman and FTIR spectroscopy have significant potential for bacterial identification (including resistance-detection) [10,13,14,16,17,18,19] in the context of clinical applications. However, the vast majority of reports available in the literature still require preculturing (including in the presence of antibiotics). In such cases, it is important to limit the preculturing time to a minimum, often falling within the lag phase. For such applications, knowledge regarding biochemical changes within the lag phase and associated spectral markers is critical because even small changes may hamper the outcomes of analysis. On the other hand, the biochemistry and physiology associated with intra-lag changes of bacteria and intra-lag processes are still not clear, while genotypic-based studies remain challenging due to low bacterial numbers [2]. Thus, a spectroscopy-based approach to clarifying biochemical changes within the lag phase has the potential to provide insight into its physiology and biochemistry.

Here, we present a multimodal spectroscopic approach in order to study alterations in the biochemical composition of *Staphylococcus aureus* over a timeframe of 6 h of growth in liquid medium, with particular emphasis on intra-lag differences. We probed at regular intervals of 30 min for the first 2 h of experimentation, then subsequently every 2 h (0, 30, 60, 90, 120, 240, 360 min). We collected both IR (population level) and Raman (single-cell level) spectra and complemented them by monitoring the growth using standard optical density measurements. Using principal component analysis (PCA) together with an analysis of spectral patterns and integral intensities of selected bands for single and combined modalities, we identified a range of spectra markers, reflective of inter- and intra-phase biochemical changes.

## 2. Materials and Methods

### 2.1. Bacterial Growth and Sample Preparation

For bacterial growth experiments, *S. aureus* (strain AP308) was used. An overnight culture was used to inoculate 600 mL of heart infusion (HI) medium to achieve a bacterial density of 0.2 (monitored via optical density measurements at 625 nm, OD_625_). The culture was incubated at 37 °C with aeration for 6 h. Samples (30 mL) were collected at time points of 0, 30, 60, 90, 120, 240 and 360 min. We used 1-mL samples for OD_625_ measurements, to monitor growth. The remaining volume of each sample was centrifuged at 10,000× *g* for 2 min and subsequently washed three time, using ultrapure, sterile water. After the final wash, the pallets were resuspended in water. Next, 20 µL of suspension was placed on Raman grade CaF_2_ window and air-dried. The remaining volume of samples were centrifuged (10,000× *g* for 2 min) and used directly for ATR measurements. The experiments were conducted in triplicate.

### 2.2. Raman Measurements

Data were collected using a WITec confocal CRM alpha 300 Raman microscope (WITec alpha300 R, Melbourne, Australia) equipped with a CCD detector cooled to −60 °C, 600 grooves/mm grating and an air-cooled solid-state laser operating at 532 nm. A dry Olympus MPLAN (100x/0.90NA) objective was used, with the laser coupled to the microscope by an optical fiber 50 µm in diameter. Prior to measurements, the instrument was calibrated using a silicon plate (520.5 cm^−1^). For each sample, nine technical replicates were recorded, with each spectrum from a different cell. Spectra were recorded in the spectral range of −73 to 3800 cm^−1^, with a spectral resolution of 3 cm^−1^. The integration time was set to 1 s, with 50 accumulations per spectrum.

### 2.3. Infrared Measurements

For infrared data collection, we used attenuated total reflectance—Fourier transform infrared spectroscopy (ATR-FTIR). ATR spectra were collected using a Bruker Alpha FTIR (Ettlingen, Germany) spectrometer equipped with an attenuated total reflection (ATR) sampling device that contained a single bounce diamond internal reflection element and a KBr beam splitter with a globar source and a deuterated triglycine sulfate detector. We transferred 0.5 µL of bacterial pallets directly onto the crystal surface and dried until no change in spectra could be seen (monitored via a live view). Spectra were collected in the range of 4000–800 cm^−1^ with a spectral resolution of 8 cm^−1^. We collected 64 and 128 scans for sample and background, respectively. Background was collected prior to spectral data collection, then regularly through the course of the experiment between every three measurements. For each sample, three technical replicates were collected, with data collection randomized on a technical replicate level. ATR correction was applied to all spectra directly after data collection using OPUS 8.0 software (Bruker, Germany).

### 2.4. Data Analysis

Data analysis was conducted with MATLAB 8.6 2015b (Mathworks, Natick, MA, USA), PLS toolbox v8.2 (Eigenvector research, Manson, IA, USA), WITec Project Plus (WITec, Germany) and Origin Pro 9.1 (Origin Lab Corp., Northampton, MA, USA). Raman spectra were preprocessed using cosmic spike removal. Second derivatives of Raman spectra were calculated using the Savitzky–Golay (SG) algorithm with 17 smoothing points. The whole dataset was then normalized using a standard normal variate (SNV). Preprocessing of ATR spectra included the calculation of second derivatives (SG, 15 smoothing points) and SNV normalization. Average spectra for each time point were obtained by averaging firstly the technical replicates and subsequently the biological replicates. The average spectra for lag and log phases were calculated as averages of the corresponding time-points (lag: 30, 60, 90, 120 min; log: 240 and 360 min). The integral intensities of bands were calculated for second derivatives, with statistical significance calculates using a T-test and ANOVA. The patterns of spectral behaviors were calculated in MATLAB using second derivatives (in-house script) and represent the change of spectra for each point relative to the previous time point or previous phase. Data were analyzed for Raman and ATR modalities independently, then in combination (multimodal approach). Multimodal PCA was conducted after mean centering using spectral ranges of 1800–900 cm^−1^ (ATR) and 1800–600 cm^−1^ (Raman).

## 3. Results and Discussion

### 3.1. Growth Phases over the Course of the Experiment

We first determined the growth phases of bacteria observed over the 6 h course of the experiment. Figure 1 shows the growth curve for the *S. aureus* strain based on optical density measurements. It is evident that within the timeframe of the experiment (360 min) bacteria went through the lag phase (0–120 min) then entered an exponential (log) phase (>120 min) of growth. This is consistent with literature reports [28,29].

### 3.2. Inter-Phase Growth Changes

We first focused on a comparison between average spectra of bacteria in lag (30–120 min) and log (>120 min) phases. The average ATR and RS spectra, together with an analysis of spectral patterns and relative intensities, are shown in Figure 2. Major differences can be seen in both ATR (Figure 2A) and RS (Figure 2B) spectra between growth phases.

In ATR (Figure 2A) these differences were localized predominantly within the low wavenumber region (1250–900 cm^−1^) and expressed through bands at 1215, 1117, 1082, 1057, 1032, 994 and 965 cm^−1^. A detail assignment of these is given in Table 1. Notably, each was predominantly associated with nucleic acids (1215, 1082, 1057, 965 cm^−1^) and carbohydrates (1032 cm^−1^). These clearly increased during the lag phase compared to the log phase (Figure 2C,E), which was accompanied by a prominent decrease in the relative intensity of the band at 1394 (δ(CH_3_)) together with a decrease of Amide II intensity (relative to Amide I), indicating a change in protein composition (Figure 2C,E).

The Raman spectra (Figure 2B) also showed substantial differences between bacteria in lag and log phases, with multiple bands showing increased relative intensity in the lag phase. This is particularly pronounced in the bands at 1575, 1480 and 1244 cm^−1^, all of which are attributed to nucleic acids (Figure 2D,F). Similar changes, although less pronounced, can be seen in other nucleic acid bands at 785 and 729 cm^−1^. A detailed assignment of the bands is provided in Table 1. At the same time, the intensity of the band at 1453 cm^−1^ (δ(CH_2_), proteins) decreased in the log phase relative to lag phase (Figure 2D,F). These results maintain a good correlation with the results of the ATR analysis. The differences in Raman spectra between bacteria in different growth phases were previously reported [24,25]. In particular, a decrease in Raman bands related to nucleic acids (1484, 783 cm^−1^) in the log phase compared to lag phase was observed [25]. This were attributed to changes in relative nucleic acids and protein content between various growth phases [24].

### 3.3. Intra-Phase Growth Changes

We further focused on differences observed between spectra of bacteria within the lag phase (30–120 min.). The second derivatives of the average IR and Raman spectra from each time point are given in Figure 3. As can be seen, the intra-lag differences (Figure 3) are less pronounced than the inter-phase ones but are still prominent. In the FTIR spectra (Figure 3A) these changes in the log and lag phase are mostly observed in bands located between 1100–1000 cm^−1^ (Figure 3B), predominantly through the band at 1032 cm^−1^. In Raman spectra (Figure 3C), the major variability can be noticed in the spectral ranges of 1140–960 cm^−1^ (Figure 3D) and 1510–1425 cm^−1^ (Figure 3E) through bands at 1098 and 1045 cm^−1^ together with varying relative intensities of bands at 1480 and 1453 cm^−1^.

Spectra recorded at the beginning of the experiment (time point 0) were not included in the general comparison of average spectra from intra-lag bacteria for clarity of presentation. The experiments were initiated using an overnight culture. In such culture, cells were no longer in the lag phase, but rather in the log or even stationary phase. Indeed, spectra recorded for bacteria at the beginning of the experiment resembled bacteria in the log phase to a much greater extent than bacteria in the lag phase (Appendix A). Thus, the difference between spectra recorded for bacteria harvested at the beginning of the experiment and bacteria harvested throughout the lag phase were much more pronounced compared to intra-lag spectral differences.

To further elucidate the observed intra-lag spectral variability, we performed a detailed analysis of the patterns of changes for IR and Raman spectra recorded over the growth of the *S. aureus* (Figure 4). Figure 4A,B presents full patterns of spectral changes across all wavenumber values for IR and Raman spectra, in the ranges of 1800–900 cm^−1^ and 1800–600 cm^−1^, respectively. Each row corresponds with consecutive time points and demonstrates the change of the intensity at every wavenumber relative to the previous time point. These patterns indicate the bands constituting the main source of variability for each time point (relative to previous), along with the time point of the most prominent changes. As expected, both IR and Raman spectra indicated that by far the largest extent of differences existed between the 0 and 30 min time points. This clearly reflects the multitude of biochemical processes occurring upon the adaptation of bacteria from the overnight culture to ‘new’ conditions. At the same time, it indicates that the vast majority of these adaptive processes occur in fact within the first 30 min.

Several bands showing large variability in the analysis of patterns of spectral changes were further selected and their relative integral intensities were calculated together with their statistical significance (Figure 4C,D). From the FTIR data it is clear that an increase in the content of nucleic acids (visible via bands at 1215, 1082 and 964 cm^−1^) progressed significantly from the beginning, reaching its maximum at 60 (964 cm^−1^) or 90 min (1082, 1215 cm^−1^). Of note is the fact that not all nucleic acid-associated bands reached their maximum relative intensity at the same time point. This may have been a result of the contribution of vibrational modes from other compounds to individual bands, or it could reflect ongoing structural changes in nucleic acids. In addition, a decrease in the relative intensity of the band at 1394 cm^−1^ and amide II is visible, with the major changes occurring within the first 30 min and further progressing slightly until 60 min. Finally, a significant change can be observed in the relative intensity of the band at 1035 cm^−1^, with a very prominent increase after 30 min followed by a further increase after 60 min and a rapid drop after 90 min. The pattern of intensity changes in this band differs from the one for nucleic acids and proteins, indicating the likelihood of a different origin—most probably carbohydrate.

The outcomes of the analysis of Raman data (Figure 4D) remained in good correlation with the outcomes of the FTIR data analysis. The nucleic acid-associated bands increased their relative intensity within the lag phase, reaching a maximum after 60 min (785 cm^−1^) or 90 min (1483 cm^−1^). A rapid decrease in the relative intensity of the band at 1453 cm^−1^ is visible after the first 30 min. This relative decrease may have resulted from a decrease in protein and lipid content, or it might reflect an increase in nucleic acid content. A prominent difference is also visible in the band at 1045 cm^−1^, which is typically associated with carbohydrates. This band rapidly increases within the first 30 min of the lag phase and then steadily decreases over the next 2 h.

Multimodal PCA (Figure 5) performed on combined IR and Raman datasets further confirmed the clear-cut differences between spectra of bacteria in lag phase collected over 30 min intervals. As can be seen, a time-dependent discrimination was achieved, with the spectral differences consistent between independent biological replicates of the experiment. The first PC, explaining 48.91% of variability in dataset, is clearly dominated by nucleic acid bands both in the IR and Raman parts. The nucleic acid bands (in IR at 964, 1082 and 1215 cm^−1^ and in Raman at 1483 cm^−1^) are more intense in the spectra of bacterial pellets collected between 60 and 120 min. The pellets collected at the beginning (0 min) and after 30 min of the culture were associated with higher relative content of proteins, visible particularly in the IR through bands at 1540, 1515 and 1394 cm^−1^. The decrease of the band at 1459 cm^−1^ could potentially reflect reduced/changed protein content, or it could be a secondary-effect of increased nucleic acid content. Nevertheless, the Raman part of loading 1 clearly indicates that out of the plethora of nucleic acid-associated bands in the Raman spectra, the ratio of 1483/1459 cm^−1^ appears to have been the most significant probe of relative nucleic acid content. Loading 2 clearly separates the spectra of intra-lag bacterial collected after 30 and 60 min from the others. Interestingly, this separation is driven in the Raman part of the loading by one band located at 1047 cm^−1^ and assigned to carbohydrates. This band is positively correlated with the IR band at 1035 cm^−1^ in loading 2, also attributed to carbohydrates. The IR part of loading 2 further reveals the contribution of one nucleic acid band (1082 cm^−1^) and one protein band (1550 cm^−1^) towards the discrimination. However, unlike in loading 1, the correlation between nucleic acid and protein bands for loading 2 is positive. Both are negatively correlated with carbohydrate-related bands.

The presented results clearly demonstrate the utility of both IR and RS in probing the chemical changes of bacteria, even within the same growth phase. The changes observed in this study reflect the cellular activity that may be associated with adaptation to new environments and in preparation for division. However, it should also be highlighted that these changes may have been affected by continuous probing of the culture, which resulted in a steady decrease of the culture volume. Such decreases affect the nutrient availability and metabolic activity of cells.

## 4. Conclusions

We demonstrated that multimodal vibrational spectroscopy is a powerful tool to probe the biochemical composition of bacteria and is sensitive to even minor alterations in composition. Both FTIR and Raman spectroscopic methods detected significant changes in the chemical composition of bacteria in different growth phases. For the lag and log phases these changes were related predominantly to different relative nucleic acid contents, and were accompanied by alterations in protein composition. Moreover, for the first time we demonstrated changes in the biochemical compositions of bacteria within the lag phase probed at regular 30 min intervals. We conducted an analysis of spectral patterns and relative intensities together with multimodal PCA, performed simultaneously on ATR and Raman data. This enabled us to achieve a clear-cut, time-dependent discrimination between intra-lag bacteria. The dominant spectral differences were associated with relative nucleic acid content, which reached its highest level after 60 and 90 min. This was expressed by bands at 1215, 1085 and 965 cm^−1^ in ATR and at 1483 cm^−1^ in Raman. Further to that, we observed an alteration in the protein composition (Amide II/Amide I) and substantial changes in relative carbohydrate content, visible via bands at 1035 cm^−1^ (ATR) and 1045 cm^−1^ (Raman). These changes may reflect the cellular activity aimed at adaptation to a new environment or in preparation for division. Altogether, our results demonstrate the possibilities offered by multimodal vibrational spectroscopies towards providing a biochemical characterization, enabling the study of microbial physiology even given low bacterial numbers. Such biochemical probing opens a new door towards studying lag-phase related events.

## Figures and Tables

**Figure 1 sensors-20-03452-f001:**
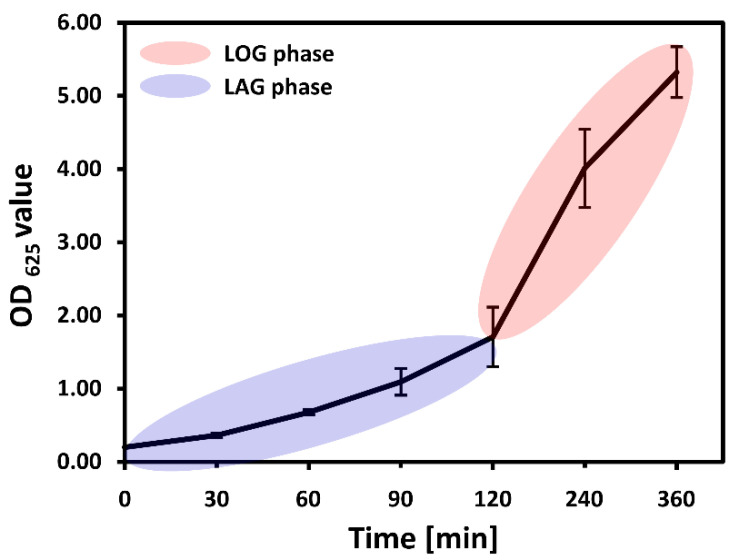
Bacterial growth curve for *Staphylococcus aureus* (AP308) grown in HI medium, monitored by optical density at 625 nm. The timeframe of the experiment was 6 h, with an initial OD of 0.2, further probed after 30, 60, 90, 120, 240 and 360 min. The blue and red ellipses mark lag and log phase, respectively.

**Figure 2 sensors-20-03452-f002:**
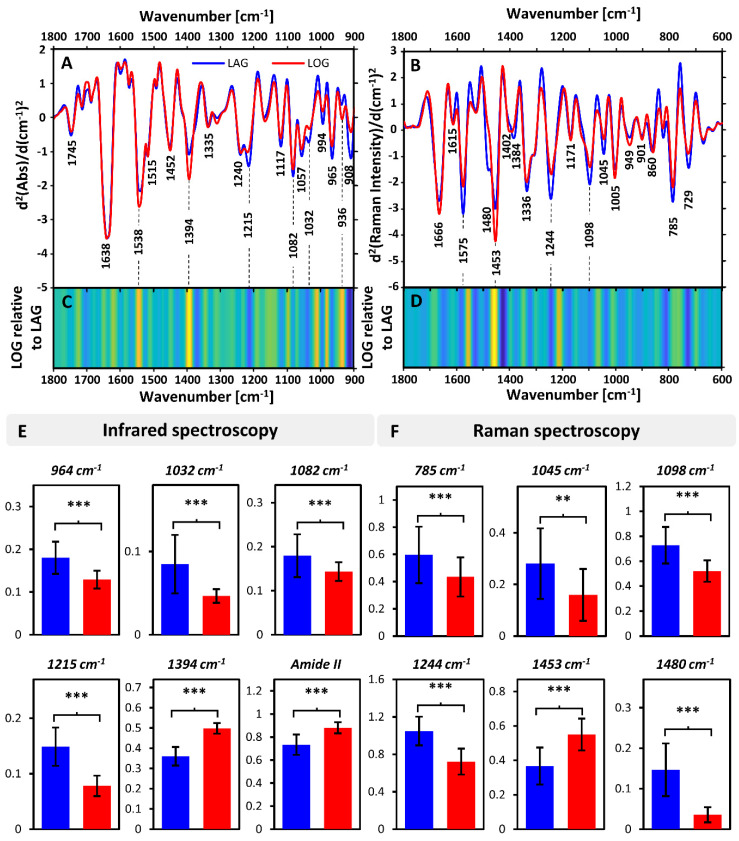
Comparison of IR (ATR) and Raman spectra recorded for bacteria in lag and log phases. Average second derivatives of: (**A**) ATR spectra in the range of 1800–900 cm^−1^ and (**B**) Raman spectra in the range of 1800–600 cm^−1^, together with marked prominent bands. Analysis of patterns of changes between second derivatives of spectra of bacteria in the lag relative to log phases for (**C**) ATR and (**D**) Raman data. Integral intensities of bands calculated for selected bands in (**E**) ATR and (**F**) Raman spectra, based on second derivatives. The integral intensities are given relative to the integral intensity of Amide I. *** denotes significance of *p* < 0.01 and ** of *p* < 0.05. The color-coding is given in (**A**) and kept consistent throughout whole figure.

**Figure 3 sensors-20-03452-f003:**
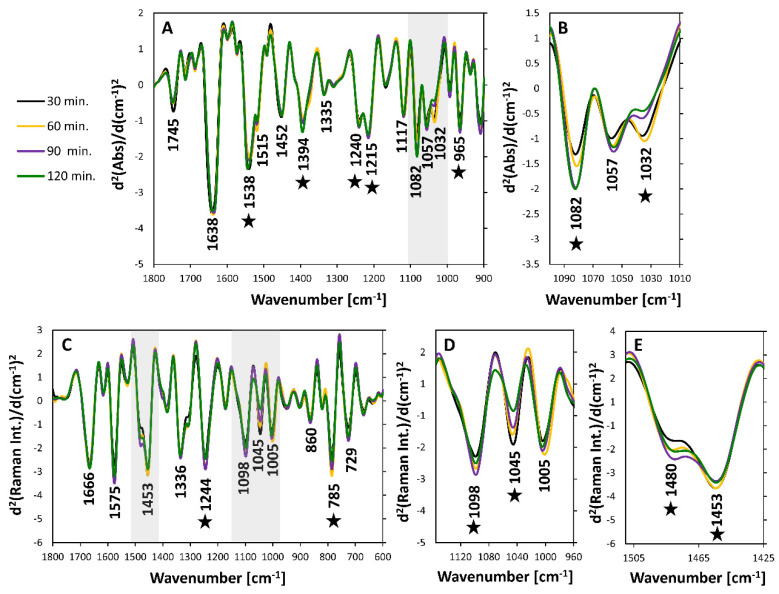
Comparison of ATR and Raman spectra recorded for bacteria within the lag phase. (**A**) Average second derivatives of ATR spectra in the range of 1800–900 cm^−1^ with marked prominent bands together with (**B**) magnification of the spectral range of 1100–1010 cm^−1^. (**C**) Average second derivatives of Raman spectra in the range of 1800–600 cm^−1^, together with magnification of spectral ranges (**D**) 1140–960 cm^−1^ and (**E**) 1510–1425 cm^−1^. Stars mark bands are showing the most prominent changes.

**Figure 4 sensors-20-03452-f004:**
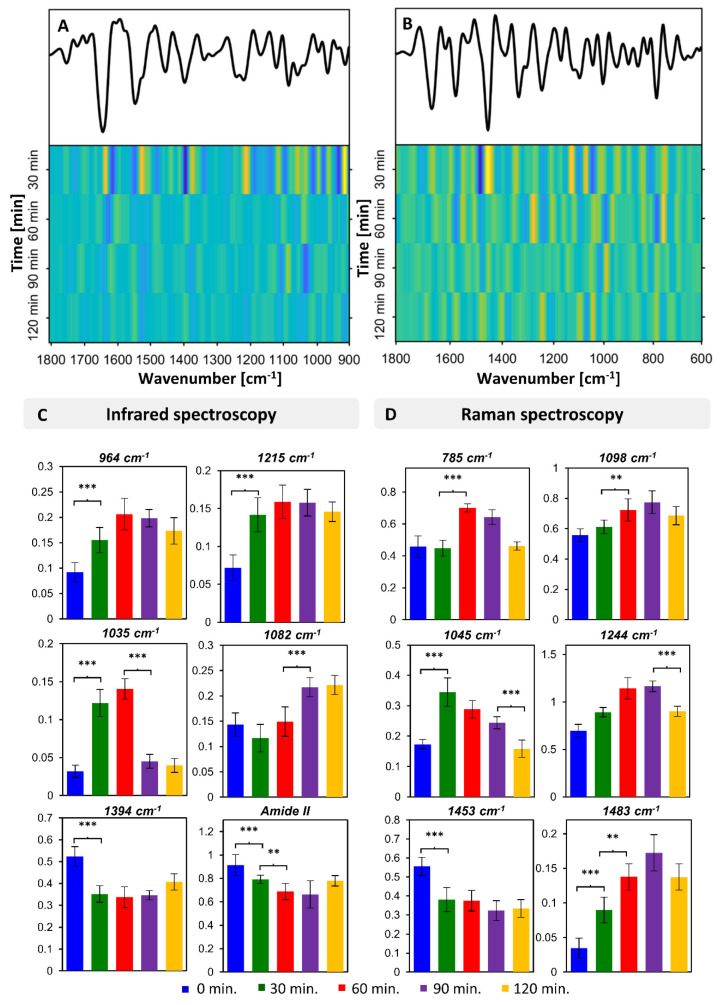
Spectral patterns of chemical changes in *S. aureus* during growth. Analysis of spectral changes over time for *S. aureus* for ATR-FTIR (**A**,**C**) and Raman spectra (**B**,**D**). (**A**,**B**) Panels presenting the change at each wavenumber in the range: ((**A**) ATR-FTIR) 1800–900 cm^−1^ and ((**B**) Raman) 1800–600 cm^−1^, at each time point, relative to the previous time point. (**C**,**D**) Bar charts presenting relative changes in intensity of selected bands in (**C**) ATR-FTIR and (**D**) Raman spectra, with the bands specified above each bar chart. *** marks statistical significance of *p* < 0.01 and ** of *p* < 0.05.

**Figure 5 sensors-20-03452-f005:**
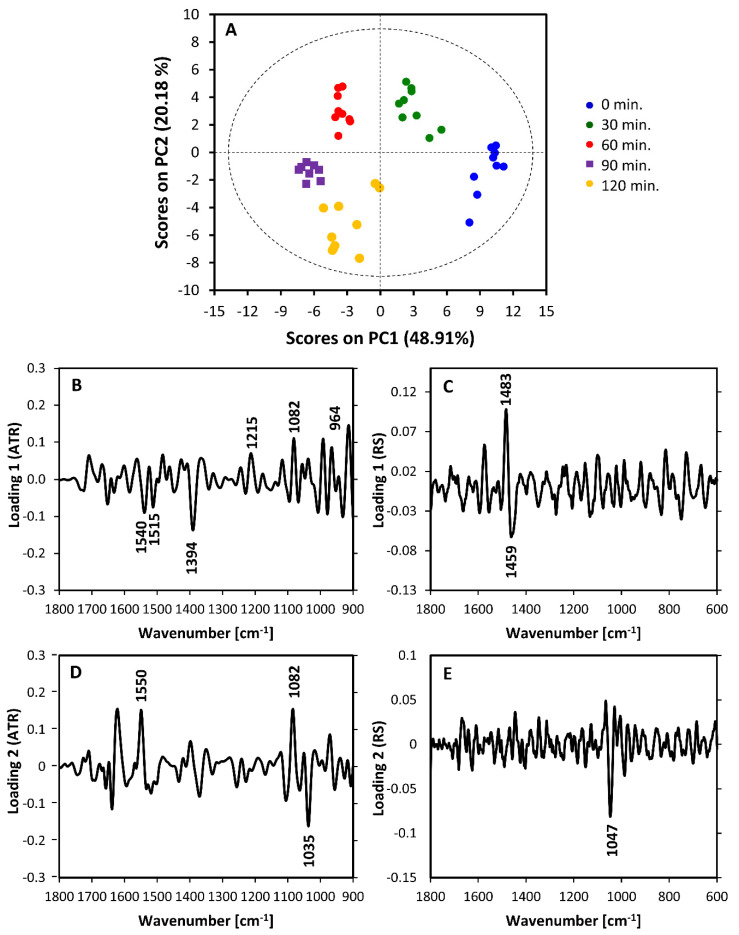
Multimodal PCA of intra-lag changes in *S. aureus,* performed in combined spectral ranges of 1800–900 cm^−1^ (ATR-FTIR) and 1800–600 cm^−1^ (Raman). (**A**) Scores plot (PC1 vs PC2) together with corresponding loadings: (**B**,**C**) loading 1 and (**D**,**E**) loading 2. The IR parts of the loadings are given in (**B**,**D**) and the Raman parts are presented in (**C**,**E**).

**Table 1 sensors-20-03452-t001:** Assignment of infrared (IR) and Raman bands observed in spectra of *S. aureus* over 6 h growth.

IR Spectroscopy			Raman Spectroscopy		
Band Position [cm^−1^]	Assignment (Vibration)	Assignment (Compounds)	Band Position [cm^−1^]	Assignment (Vibration)	Assignment (Compounds)
1745	ν(C = O)	Lipids	1666	ν(C = C), Amide I	Lipids, Proteins
1638	Amide I	Proteins (β-sheet)	1615	ν(C = C),	Tyr, Trp
1538	Amide II	Proteins (β-sheet)	1575	Ring breathing (guanine, adenine)	Nucleic acids
1515		Tyrosine	1480	Ring breathing (guanine, adenine)	Nucleic acids
1452	δ_as_(CH_3_), δ(CH_2_), δ(CH)^−^	Proteins, Lipids, Polysaccharides	1453	δ(CH_2_), δ(CH_3_)	Protein
1394	δ_s_(CH_3_)	Proteins	1402	δ(CH_3_)	Protein
1335	δ(CH), ring	Polysaccharides	1384		
1240	ν_as_(PO_2_)^−^ Amide III	Nucleic acids, Phospholipids, Proteins	1336	Amide II, ring breathing (A,G)	Protein, Nucleic acids
1215	ν_as_(PO_2_)^−^ Amide III	Nucleic acids, Phospholipids, Proteins	1244	Amide III	Protein
1117	ν(C-C), ν(C-O), ν(P-O-C),	Polysaccharides, RNA	1171	δ(C-H)	Phe, Tyr
1082	ν_s_(PO_2_)^−^	Nucleic acids, Phospholipids	1098	(PO_2_)^−^	Nucleic acids
1057	ν(C-O-C)		1045	ν(C-C), ν(C-C)	Polysaccharides
1032	ν(C-C), ν(C-O),	Polysaccharides	1005	ν(C-C)	Phenylalanine
994	ν(C-C), ν(C-C)	Polysaccharides, ribose	785	O-P-O, ring breathing	Nucleic acids
965	ν(C-C), ν(C-C)	DNA	729	Ring breathing, A	Nucleic acids

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
