# Peer review of "Vibrational Spectroscopy as a Sensitive Probe for the Chemistry of Intra-Phase Bacterial Growth"

_sensors, 2020, doi:10.3390/s20123452_

Round 1

Reviewer 1 Report

The paper „Vibrational spectroscopy as a sensor for intra-phase bacterial growth” is about to use FTIR and Raman spectroscopy to detect changes in the chemical composition of bacteria within the same phase. The structure of this paper is very good, but I have some comments to the manuscript:

  1. In the Abstract more information about wavenumbers which are spectroscopic markers should be added.
  2. In the introduction author should write about other researches where FTIR an Raman spectroscopy were used in microbiological research e.g. International journal of molecular sciences 20 (4), 988 (2019).
  3. Authors write that they used 17 smoothing points. It is very large number, which can be responsible for vanishing spectral information.
  4. In the manuscript average of obtained spectra or selected spectra were visible?

I recommend major revision of this manuscript.

Reviewer 2 Report

Manuscript ID: Sensors-823775

The study by Kochan et al is aimed at using FTIR (ATR) combined with Raman spectroscopy to determine the biochemical changes in S. aureus at intra-phase level, with an emphasis on lag phase. The study has reported some interesting findings, and its also nice to see that the authors have attempted to assign the significant vibrational bands and link the observed spectral changes to potential biochemical activities. However, there are some fundamental questions (outlined below) regarding the study design and sample preparation that needs attention. Therefore, I believe that the study in its current form is not fit for publication and suggest major revision.

Major comments

  • In the methods section it is stated that the starting volume of the batch cultures were 600 mL, and samples (90 mL) were taken at various timepoints. This means by the time the 120 min samples are collected the culture volume has been reduced by over 70% (450 mL) of the starting volume which will affect nutrient availability, and of course the metabolic activity of the cells. This is a major flaw in the experimental design, unless the authors have missed out other supporting information that proves to be otherwise.
  • The medium that was used in this study is heart infusion (HI) broth, which is considered a rich medium and hence the bacteria will undergo various metabolic shifts depending on their preference for the available nutrients. Therefore, to achieve a generalised understanding of the intra-phase biochemical changes, it would have been more appropriate to conduct the experiment using a variety of popular rich media that would allow for a more robust conclusion to be made.

Minor comments

  • Comparison of the Raman single cell spectral data with the FTIR data is not appropriate, as Raman will capture the heterogeneity of the community at single cell level, while such information are averaged and lost in the FTIR data due to the lower spatial resolution of this technique.

Reviewer 3 Report

The paper "Vibrational spectroscopy as a sensor for intra-phase bacterial growth" reports a multimodal approach to probe changes of chemical composition of bacteria during the same growth phase.
The metodology combines Raman and FTIR spectra and uses several data analysis including Principal Component Analysis to achieve time-dependent discrimination of intensity changes of bands originating from nucleic acids, proteins and carbohydrate.
The paper is well written and introduces sounding results on spectral changes observed for the staphylococcus aureus.
Based on this, I recommand its publication after minor changes.

Minor corrections:
Line 131 : was than normalized -> was then normalized
Line 163 : This is accopmanied by -> This is accompanied by
Line 165 : Amide II intesity -> Amide II intensity
Line 172 : in good correlataion with -> in good correlation with

Round 2

Reviewer 2 Report

No further comments.